# Enhancement of Human Immunodeficiency Virus-Specific CD8^+^ T Cell Responses with TIGIT Blockade Involves Trogocytosis

**DOI:** 10.3390/pathogens13121137

**Published:** 2024-12-23

**Authors:** Nazanin Ghasemi, Kayla A. Holder, Danielle P. Ings, Michael D. Grant

**Affiliations:** 1Immunology and Infectious Diseases Program, Division of BioMedical Sciences, Faculty of Medicine, Memorial University of Newfoundland, St. John’s, NL A1B 3V6, Canada; nghasemi@mun.ca (N.G.); kayla.holder@mun.ca (K.A.H.); dings@mun.ca (D.P.I.); 2Department of Biomedical Informatics, University of Colorado School of Medicine, Denver, CO 80045, USA; 3Department of Immunology and Microbiology, University of Colorado School of Medicine, Denver, CO 80045, USA

**Keywords:** HIV, TIGIT, CD8^+^ T cell, cytotoxicity, trogocytosis

## Abstract

Natural killer (NK) and CD8^+^ T cell function is compromised in human immunodeficiency virus type 1 (HIV-1) infection by increased expression of inhibitory receptors such as TIGIT (T cell immunoreceptor with Ig and ITIM domains). Blocking inhibitory receptors or their ligands with monoclonal antibodies (mAb) has potential to improve antiviral immunity in general and facilitate HIV eradication strategies. We assessed the impact of TIGIT engagement and blockade on cytotoxicity, degranulation, and interferon-gamma (IFN-γ) production by CD8^+^ T cells from persons living with HIV (PLWH). The effect of TIGIT engagement on non-specific anti-CD3-redirected cytotoxicity was assessed in redirected cytotoxicity assays, and the effect of TIGIT blockade on HIV-specific CD8^+^ T cell responses was assessed by flow cytometry. In 14/19 cases where peripheral blood mononuclear cells (PBMC) mediated >10% redirected cytotoxicity, TIGIT engagement reduced the level of cytotoxicity to <90% of control values. We selected PLWH with >1000 HIV Gag or Nef-specific IFN-γ spot forming cells per million PBMC to quantify the effects of TIGIT blockade on HIV-specific CD8^+^ T cell responses by flow cytometry. Cell surface TIGIT expression decreased on CD8^+^ T cells from 23/40 PLWH following TIGIT blockade and this loss was associated with increased anti-TIGIT mAb fluorescence on monocytes. In total, 6 of these 23 PLWH had enhanced HIV-specific CD8^+^ T cell degranulation and IFN-γ production with TIGIT blockade, compared to 0/17 with no decrease in cell surface TIGIT expression. Reduced CD8^+^ T cell TIGIT expression with TIGIT blockade involved trogocytosis by circulating monocytes, suggesting that an effector monocyte population and intact fragment crystallizable (Fc) functions are required for mAb-based TIGIT blockade to effectively enhance HIV-specific CD8^+^ T cell responses.

## 1. Introduction

Effective antiretroviral therapy (ART) has transformed human immunodeficiency virus (HIV) infection from terminal disease into chronic manageable condition, but persons living with HIV (PLWH) sustain immunological abnormalities and remain prone to non-acquired immunodeficiency syndrome (AIDS)-related illness [1,2,3]. Two key issues affecting PLWH are the residual inflammation presumed to drive accelerated onset and increased prevalence of age-related morbidities and the pervasive antiviral effector cell dysfunction that poses a significant challenge to the efficacy of immunotherapeutic strategies [4,5,6,7]. Natural killer (NK) and CD8^+^ T cell defects in PLWH include incomplete maturation, abnormal accumulation of inhibitory receptors, functional exhaustion, and terminal senescence [5,6,8,9,10,11,12]. Invigoration of effector cells by “checkpoint inhibition”, as employed in cancer therapy, could address some of these defects, improve general immune function, and ultimately support HIV cure strategies [13,14].

Many inhibitory receptors depress NK and CD8^+^ T cell effector functions upon engaging their ligand [12]. Consequently, several of these receptors are targeted by monoclonal antibody (mAb)-based drugs already licensed for cancer treatment and others are investigational targets in ongoing clinical trials. The striking success of mAb-based blockade in invigorating immune effector cells for cancer treatment has encouraged diversification of the checkpoint inhibitor approach across new applications, including chronic infection [14]. The inhibitory receptor T cell immunoreceptor and immunoglobulin with tyrosine-based inhibitory motif (ITIM) domains (TIGIT) is an attractive checkpoint target in the context of HIV infection for the following reasons: (1) TIGIT expression increases over the course of HIV infection on CD4^+^ T cells, CD8^+^ T cells and NK cells, often to a point where the majority of CD8^+^ T cells and NK cells express TIGIT; (2) virtually all HIV-specific CD8^+^ T cells express TIGIT; (3) the poliovirus receptor (PVR) that TIGIT engages to mediate inhibition is upregulated on CD4^+^ T cells in PLWH, especially on follicular helper T cells (T_FH_) in which the HIV reservoir is concentrated; (4) reactivation of HIV in primary CD4^+^ T cells triggers PVR expression; (5) DNAM-1 and TACTILE, TIGIT antagonists that also bind PVR, are downregulated in HIV infection, compounding the impact of TIGIT overexpression [15,16,17,18,19,20,21,22]. These findings suggest that TIGIT inhibition of CD8^+^ T and NK cell surveillance against HIV-infected CD4^+^ T cells and monocytes contributes to immune dysfunction in chronic HIV infection and that TIGIT blockade could enhance antiviral effector cell function, in general, and especially against HIV [23]. Responses to TIGIT blockade with in vitro and in vivo studies have been variable with no clear consensus on the extent to which PVR/TIGIT interactions impact effector cell functions across different forms of stimulation and different types of target cell engagement, nor on the importance of intact Fc receptor function of the mAb used for therapeutic efficacy [15,16,22,24,25,26,27,28]. We previously showed that TIGIT blockade increased NK cell-mediated killing of PVR-expressing targets and accelerated NK cell-mediated elimination of autologous CD4^+^ T cells in which endogenous HIV replication was stimulated [21]. Responses to TIGIT blockade in this previous study were greater in persons co-infected with cytomegalovirus (CMV) and increased with the level of TIGIT expression on NK cells [21]. The aims of this current study were to determine whether TIGIT engagement on circulating CD8^+^ T cells from PLWH reduces cytotoxic T cell-mediated cytotoxicity and to define host features associated with HIV-specific CD8^+^ T cell responsiveness that could help identify persons most likely to benefit from TIGIT blockade. Our data support the potential utility of TIGIT blockade to address antiviral effector cell dysfunction in PLWH and affirm the individual variability in responsiveness. Consideration of the environmental context within which TIGIT blockade is enacted and the phenotypic status of the cells targeted for TIGIT blockade will be beneficial in selecting candidates for treatment.

## 2. Materials and Methods

### 2.1. Sample Collection and Processing

This study conformed to recommendations of the Canadian Tri-Council Policy Statement: Ethical Conduct for Research Involving Humans and ethical approval was received from the Health Research Ethics Authority of Newfoundland and Labrador. In accordance with the Declaration of Helsinki, written informed consent for participation was obtained by the provincial immunodeficiency nurse at the Newfoundland and Labrador Provincial Immunodeficiency Clinic, St. John’s, NL, Canada. Whole blood was collected by forearm venipuncture into sterile vacutainers containing acid citrate dextrose (ACD) anticoagulant. Peripheral blood mononuclear cells (PBMC) were isolated from whole blood by density gradient separation using Ficoll-Paque PLUS density gradient media (Cytiva HyClone^TM^, Logan, UT, USA) following the Canadian Autoimmunity Standardization Core consensus standard operating procedure (version: 21 March 2019; https://www.bcchr.ca/CAN-ASC/protocols, accessed on 28 November 2019). Briefly, whole blood was centrifuged at 500× *g* for 10 min in a Beckman Coulter Allegra X-12 R centrifuge. The upper plasma layer was removed, aliquotted into 1.5 mL Eppendorf microtubes, and stored at −80 °C until use. The lower cellular layer was diluted with phosphate-buffered saline (PBS, Sigma Aldrich, St. Louis, MO, USA) to the original whole blood volume, carefully underlaid with Ficoll-Paque and centrifuged at 400× *g* for 30 min without braking. The interface layer containing mononuclear cells was harvested, washed with PBS containing 2% fetal bovine serum (FBS, HyClone^TM^, Thermo Fisher Scientific, Waltham, MA, USA), and centrifuged for 10 min at 300× *g*. The supernatant was discarded, pellet resuspended in 10 mL 2% FBS in PBS, and platelets removed by discarding the supernatant after 140× *g* centrifugation for 10 min. After centrifugation, PBMC were resuspended in lymphocyte medium (Roswell Park Memorial Institute (RPMI)-1640, supplemented with 10% FCS), 200 IU/mL penicillin/streptomycin, 1% L-glutamine (all from Thermo Fisher Scientific), 1% 4-(2-hydroxyethyl)-1-piperazineethanesulfonic acid, and 0.1% ß-mercaptoethanol (Sigma-Aldrich) and counted immediately using trypan blue exclusion stain. If not used immediately, PBMC were pelleted and resuspended in freezing medium (FBS + 10% dimethyl sulfoxide (DMSO, Sigma Aldrich)), dispensed into cryovials at ~2 × 10^7^ PBMC/mL, cooled slowly overnight to −80 °C, then transferred into liquid nitrogen (LN_2_) until use. To recover PBMC for experiments, cryovials were removed from LN_2_, and rapidly thawed at 37 °C. Cells were immediately transferred into a 15 mL centrifuge tube containing 9 mL lymphocyte medium, centrifuged at 450× *g* for 5 min, resuspended in fresh lymphocyte medium, incubated at 37 °C with 5% CO_2_ overnight and counted using trypan blue exclusion dye before use.

### 2.2. Redirected Cytotoxicity Assay

Non-specific anti-CD3-triggered cytotoxicity was measured by Chromium-51 (^51^Cr) release using PBMC as the source of cytotoxic T cells (CTL). The fragment crystallizable (Fc)R-expressing P815 murine mastocytoma target cells (ATCC TIB-64TM) were labeled in a minimum volume of medium containing 100 μCi Na_2_^51^CrO_4_ (Perkin Elmer, Akron, OH, USA) for 90 min at 37 °C, 5% CO_2_. Labeled target cells were washed 4 times with PBS supplemented with 1% FBS and resuspended in lymphocyte medium at 1 × 10^5^/mL. During this time, 0.25 × 10^6^ effector cells were distributed into individual wells of a 96-well round bottom plate (Thermo Fisher Scientific) and IgG_1_ isotype control (clone 11711, R&D Systems, Minneapolis, MN, USA) or anti-TIGIT mAb (clone MBSA43, Thermo Fisher Scientific) added for a final concentration of 5 μg/mL to precoat effector cells for 30 min at 37 °C and 5% CO_2_. Target P815 cells (5 × 10^3^) were added to wells containing effector cells for a final effector to target (E:T) ratio of 50:1 with 0.125 μg/mL anti-CD3 mAb OKT3 (Thermo Fisher Scientific) in a final volume of 300 μL to trigger CTL. The lowest concentration of anti-CD3 required to trigger maximal target cell lysis was chosen to minimize competition for FcR binding. Controls for spontaneous release (targets in lymphocyte medium) and maximum release (targets with 1 N hydrochloric acid) were included. After incubation for 5 h at 37 °C, 5% CO_2_, 125 μL of supernatant was transferred to Kimble tubes (Thermo Fisher Scientific) containing 50 μL bleach. Release of ^51^Cr was measured using the Wallac 1480 Gamma Counter and specific lysis of P815 cells calculated using the following formula: Percent specific lysis = [(Experimental Release − Spontaneous Release)/(Maximum Release − Spontaneous Release)] × 100.

### 2.3. Peptide Preparation for Antigen-Specific T Cell Stimulation

To measure antigen-specific T cell activation, PBMC were stimulated with pools of overlapping synthetic peptides spanning HIV Gag and/or Nef proteins. The HIV-1 consensus clade B Nef peptide pool and the HIV-1 consensus clade B Gag peptide pool consisting of 15 mers with 11 amino acid (aa) overlap were provided by the National Institutes of Health (NIH) HIV Reagent Program. Lyophilized peptide pools were dissolved at 10 mg/mL in DMSO then diluted to 100 μg/mL with serum free RPMI and stored at −80 °C until use.

### 2.4. ELISpot Assay

After initially testing PLWH samples as they were received, we adopted an enzyme-linked immunospot (ELISpot) screening approach to test for donors with strong T cell interferon-gamma (IFN)-γ responses against HIV Gag or Nef peptide pools. Pre-coated anti-IFN-γ antibody Immunospot^TM^ plates (Cellular Technologies Limited (CTL), Shaker Heights, OH, USA)) were washed once with 150 μL PBS, then 1 μg/mL of Gag or Nef peptides or DMSO vehicle control or 1 µg/mL anti-CD3 (OKT3) positive control were added to individual wells in duplicate in 100 μL CTL medium. PBMC were added at 2 × 10^5^/well for a final volume of 200 μL and incubated 18 h at 37 °C and 5% CO_2_. Plates were washed twice with 200 μL/well PBS then twice with 200 μL/well of PBS + 0.05% Tween-20 (Sigma Aldrich). Anti-human IFN-γ detection antibody was added to each well as per manufacturer’s instructions then incubated for two hours at room temperature. ELISpot plates were washed three times with 200 μL/well of PBS + 0.05% Tween-20 and developed as per manufacturer’s instructions. The reaction was stopped by gently washing the plate with tap water and plates were air-dried face down overnight. Plates were scanned using a CTL-Immunospot Analyzer (Cellular Technologies Limited, Shaker Heights, OH, USA) and the frequency of spot forming cells (SFU)/10^6^ PBMC was calculated.

### 2.5. Assessment of HIV-Specific CD8^+^ T Cell Responses by Flow Cytometry

To evaluate the role of TIGIT in regulating HIV-specific CD8^+^ T cell responses, 2 × 10^6^ PBMC were stimulated in 1 mL lymphocyte medium with HIV peptides in the presence of 5 μg/mL TIGIT-blocking mAb (clone: MBSA43, Thermo Fisher Scientific) or an IgG_1_ isotype control (clone: 11711, R&D Systems). Following 20 min Ab pretreatment, Gag or Nef peptide pools were added at a final concentration of 1 μg/mL. An equivalent amount of DMSO was used as vehicle control. Cells were stimulated for 5 h at 37 °C and 5% CO_2_ with a final concentration of 5 μg/mL brefeldin A, 5 μg/mL monensin (Sigma Aldrich) and 5 μg/mL anti-CD107a-BV421 (clone H4A3, BioLegend, San Diego, CA, USA). After 5 h of stimulation, the cells were washed with flow buffer (5 mM EDTA, 0.5% FCS and 0.2% NaN_3_ (Sigma Aldrich) in PBS) and stained for analysis by surface and intracellular flow cytometry using anti-CD3-VioGreen (clone: REA613, Miltenyi Biotec, San Diego, CA, USA), anti-CD8-PerCP (clone: H1T8a, BioLegend), and anti-TIGIT-Alexa Fluor 647 (clone: MBSA43, Thermo Fisher Scientific) for 30 min on ice in the dark. PBMC were washed with 3 mL flow buffer and centrifuged at 300× *g* for 5 min. To stain for intracellular IFN-γ, PBMC were resuspended in 500 μL Inside Fix + Buffer (Miltenyi Biotec) as per manufacturer’s recommendations and incubated for 20 min in the dark at room temperature. Cells were washed with 3 mL flow buffer and centrifuged at 300× *g* for 5 min. Anti-IFN-γ-PE (Clone 4S.B3, Thermo Fisher Scientific) was added to the cell pellet in a final volume of 100 μL of Inside Perm (Miltenyi Biotec) and incubated 10 min in the dark at room temperature. PBMC were washed with 1 mL Inside Perm, then resuspended in flow cytometry buffer. Data were acquired on a CytoFLEX flow cytometer (Beckman Coulter, Brea, CA, USA) and analyzed using Kaluza (Beckman Coulter). Increased IFN-γ production or CD107a expression by greater than 10% of isotype control values with TIGIT blockade was considered a positive response based on the limits of sensitivity for flow cytometry.

### 2.6. Assessment of Anti-TIGIT mAb Trogocytosis

To evaluate trogocytosis mediated through Fcγ receptors (FcγR) on monocytes in the presence of anti-TIGIT mAb, we assessed TIGIT expression on CD8^+^ T cells and monocytes after 5 h incubation and compared the effect of anti-TIGIT treatment to that of the IgG_1_ isotype control. An Anti-TIGIT AlexaFluor 647 (clone: MBSA43, Invitrogen)-conjugated antibody or IgG_1_ isotype control (clone: 11711, R&D Systems) was added to 2 × 10^6^ PBMC in 1 mL lymphocyte medium for 5 h at 37 °C and 5% CO_2_. PBMC were then washed, stained for TIGIT, CD3, CD4, and CD8, as previously described, and stained with anti-CD14-VioBlue (clone: REA599, Miltenyi Biotec). Then, data were acquired using CytoFLEX (Beckman Coulter) and analyzed using Kaluza (Beckman Coulter). In time course studies, isotype control and anti-TIGIT-treated PBMC were stained, as above, 1, 3, and 5 h after incubation and analyzed. To determine whether cell to cell contact was required for the transfer of TIGIT from CD8^+^ T cells to monocytes in the presence of anti-TIGIT mAb, 2 × 10^6^ PBMC were incubated in 24-well tissue culture plates separated into equal aliquots with a semipermeable 0.4 µM Transwell insert (Corning Inc., Corning, NY, USA). PBMC placed in the insert were first exposed to fluorescence-conjugated anti-TIGIT mAb for 20 min, then washed prior to incubation. Monocytes were analyzed by flow cytometry for anti-TIGIT mAb fluorescence at time zero and from both the insert and 24-well plate after 5 h incubation at 37 °C and 5% CO_2_.

### 2.7. Statistical Analysis

Statistical analysis was carried out with GraphPad Prism software version 10, applying parametric or non-parametric testing as indicated by data distribution with two-sided *p* values < 0.05 considered significant. The normality of data distribution was assessed using the Shapiro–Wilk test and the association between variables was assessed by Spearman’s correlation.

## 3. Results

### 3.1. The Impact of TIGIT Engagement on T Cell Cytotoxicity

We and others have previously demonstrated that PLWH have higher levels of T and NK cell TIGIT expression than age and sex matched people without HIV [15,16,18,19,21]. There is considerable experimental data demonstrating that TIGIT blockade can enhance cytokine production and CD107a expression by HIV-specific and other CD8^+^ T cells [15,22,29], but there is little direct evidence that TIGIT engagement impairs human T cell-mediated cytotoxicity. Therefore, we mimicked TIGIT engagement of PVR on target cells with a mAb against TIGIT in a non-specific anti-CD3 redirected cytotoxicity assay using FcR-expressing targets (Figure 1a). Anti-CD3 was titrated to find the lowest concentration triggering maximal lysis (0.125 μg/mL) and the amount of anti-TIGIT maximizing inhibition upon engagement was also determined empirically (5.0 μg/mL). If a subject’s PBMC mediated >10% anti-CD3 redirected lysis of P815 targets, the effect of TIGIT engagement on cytotoxicity was tested in this system. Engagement of TIGIT decreased T cell killing of P815 target cells by more than 10% of the baseline level of specific lysis observed without TIGIT engagement in 74% (14/19) of the cases we tested (red lines connecting dots) and significantly lowered the mean specific lysis mediated by the CD8^+^ T cells of the group (*p* < 0.01) (Figure 1b). These data show that in the majority of cases, TIGIT engagement impairs cytotoxicity mediated by a subset of CD8^+^ T cells present in PLWH.

### 3.2. Identification of Subjects with Strong HIV-Specific T Cell Responses

The redirected cytotoxicity assay showed in a non-antigen-specific manner that TIGIT engagement reduces cytotoxicity mediated by CD8^+^ T cells from PLWH. To assess the effects of TIGIT blockade on HIV-specific CD8^+^ T cell responses and investigate features associated with responsiveness to TIGIT blockade, the responses must be robust enough to be measurable by flow cytometry. Therefore, after initial flow cytometry studies with a relatively low success rate, we decided to screen HIV Gag and Nef-specific T cell frequencies in 109 persons in our PLWH cohort with a higher throughput IFN-γ ELISpot assay before proceeding to analysis by flow cytometry. We considered 1000 SFU/10^6^ PBMC (0.1% of total PBMC) the minimum response required for inclusion in follow-up flow cytometry studies to enable acquisition of sufficient events to distinguish changes induced by TIGIT blockade over random changes in background on the instrument used. Of the 109 PLWH tested, 23 had >1000 IFN-γ producing T cells/10^6^ PBMC in response to either Gag, Nef or both following peptide stimulation (Appendix A). We selected high responders for further analysis by flow cytometry to increase our study number (n) from 17 to 40 PLWH for estimating the frequency of responsiveness to TIGIT blockade and to identify features associated with responsiveness.

### 3.3. Impact of TIGIT Blockade on CD8^+^ T Cell TIGIT Expression

The broad individual variability in responsiveness must be addressed to optimize the impact of TIGIT blockade. Response rates to checkpoint inhibition in cancer treatment are estimated at between 15% and 25% [30]. Assuming there may be a similar response rate in chronic infection, a better understanding of what features predispose towards responsiveness to TIGIT blockade would help to select those most likely to benefit from treatment and to exclude those less likely to benefit. We found in this study that in vitro exposure of PBMC to the anti-TIGIT mAb often led to a detectable level of physical loss of TIGIT expression from CD8^+^ T cells. In total, 23 of the 40 PLWH we tested lost >1/10th of their CD8^+^ T cell TIGIT expression following five-hour incubation with the anti-TIGIT mAb compared to five-hour incubation with an isotype control. General characteristics of the study group of 40 PLWH tested here are shown in Table 1 with the 23 losing CD8^+^ T cell TIGIT expression listed first. The majority of participants had plasma HIV loads below clinically detectable limits with few in each group having low levels slightly above the detection limit. While the level of TIGIT expression on CD8^+^ T cells ranged from 13 to 77% overall, the only characteristic it correlated significantly with was age (*p* = 0.11). When the association between age and level of TIGIT expression on CD8^+^ T cells was assessed separately in the responder group of 23 PLWH and the non-responder group of 17 PLWH, the correlation was only significant in the responder group (*p* = 0.0003). The 23 responders had a significantly higher mean nadir CD4^+^ T cell count than the non-responder group (410 ± 315 versus 206 ± 123, *p* = 0.0156) with no other significant differences in the characteristics available observed between groups. Our gating strategy to assess this phenomenon by flow cytometry is shown in Figure 2a–c with one representative example (Figure 2d,e). Results for the 23 individuals tested who lost >1/10th of their CD8^+^ T cell TIGIT expression following TIGIT blockade are summarized graphically in Figure 2f. Individual loss of TIGIT from the surface of CD8^+^ T cells in this group with >1/10th loss of TIGIT expression (Figure 2f) ranged from 38% TIGIT^+^ at baseline to 34% TIGIT^+^ after TIGIT blockade and 5 h incubation (4/38 = 11% loss) through 65% at baseline to 18% TIGIT^+^ after incubation (47/65 = 72% loss).

### 3.4. Impact of TIGIT Blockade on HIV-Specific CD8^+^ T Cell Responses

We compared the effects of anti-TIGIT mAb blockade to isotype control treatment on the frequency of CD8^+^ T cells producing IFN-γ or expressing CD107a in response to HIV Gag or Nef peptide pool stimulation. Our gating strategy for analysis (Figure 3a–c) and a representative example of a responder in terms of increased IFN-γ production and CD107a expression is shown (Figure 3d–f). Results are summarized for eight individuals who lost CD8^+^ T cell TIGIT expression and had increased IFN-γ production (Figure 3g) and for nine individuals who lost CD8^+^ T cell TIGIT expression with increased CD107a expression (Figure 3h) following TIGIT blockade relative to isotype control treatment. Of the 23 PLWH who responded with loss of >1/10th of their cell-surface CD8^+^ T cell TIGIT expression following TIGIT blockade, 6 showed some enhancement of both IFN-γ production and degranulation (expression of CD107a), while none of the 17 individuals with no loss of TIGIT expression showed such enhancement (Figure 3g,h). These data indicate a significantly more frequent positive response to TIGIT blockade in those subjects who lost >1/10th TIGIT expression on their CD8^+^ T cells (*p* = 0.0032, Fisher’s exact test) and suggest that loss of TIGIT expression from CD8^+^ T cells in vitro with TIGIT blockade may distinguish potential responders from non-responders in terms of enhanced antiviral effector cell function.

### 3.5. Trogocytosis of TIGIT by Peripheral Blood Monocytes

Reduction in TIGIT expression on CD8^+^ T cells with exposure to anti-TIGIT mAb was associated with a positive response to TIGIT blockade; therefore, we investigated trogocytosis as a potential mechanism underlying TIGIT loss. By gating on monocytes and evaluating anti-TIGIT mAb fluorescence levels following five-hour incubation of PBMC with anti-TIGIT mAb, we investigated whether anti-TIGIT antibody treatment resulted in monocyte uptake of anti-TIGIT mAb fluorescence. For 8 PLWH who responded to TIGIT blockade with a reduction in TIGIT expression and enhanced HIV-specific CD8^+^ T cell activation, five hours incubation of PBMC with anti-TIGIT mAb resulted in increased anti-TIGIT mAb fluorescence in the monocyte population (Figure 4a–e). Therefore, trogocytosis of TIGIT by monocytes contributes to reduction in cell-surface TIGIT expression by CD8^+^ T cells in PBMC exposed to anti-TIGIT mAb in vitro. When anti-TIGIT mAb treated PBMC were kept separated from untreated PBMC by a semi-permeable membrane for 5 h incubation, anti-TIGIT fluorescence on monocytes increased in the treated PBMC population only, illustrating that cell-to-cell contact was required for transfer of anti-TIGIT fluorescence to monocytes (Figure 4f). Harvesting and staining PBMC from two PLWH 1, 3 and 5 h into incubation with anti-TIGIT or isotype control showed a steady increase over 5 h of anti-TIGIT fluorescence in the monocyte population (Appendix A). Since trogocytosis in this system is associated with responsiveness to TIGIT blockade, the capacity of circulating monocytes to remove TIGIT by trogocytosis could be an important factor in determining T cell responsiveness to TIGIT blockade.

## 4. Discussion

Increased inhibitory receptor expression on T and NK cells in HIV infection persists through antiretroviral therapy and impairs the function of critical antiviral effector cells [15,16,18,19,21,22]. Checkpoint inhibition by inhibitory receptor blockade is a key strategy to address similar issues affecting immune responses against cancer cells [13,30]. Adaptation of checkpoint inhibition to address chronic infections could improve the general immune function of PLWH and support immunotherapies targeting the viral reservoir. We focused on the widely expressed inhibitory receptor TIGIT to investigate its regulation of different T cell antiviral effector mechanisms and potentially to identify features useful for predicting positive responses to TIGIT blockade in PLWH. Engagement of TIGIT on circulating CTL from PLWH reduced non-specific target cell killing in the majority of cases we tested, demonstrating the generalized potential for signaling through TIGIT to reduce T cell-mediated cytotoxicity. While enhancement of HIV-specific CD8^+^ T cell function by TIGIT blockade did not occur in the majority of cases we tested, the response rate closely matched that seen with in vivo checkpoint inhibition in cancer therapy [30]. This reinforces the need to define features predicting responsiveness to capitalize on directing treatment towards those most likely to benefit. In every case where TIGIT blockade enhanced HIV-specific CD8^+^ T cell function, it also reduced TIGIT expression on CD8^+^ T cells. Loss of TIGIT expression occurred at least in part through monocyte-mediated trogocytosis and, therefore, the capacity of circulating monocytes to mediate the trogocytosis of TIGIT may be one feature useful for distinguishing potential responders to TIGIT blockade from non-responders. This also depends on the Fc region of the mAb used for TIGIT blockade being competent to engage Fc receptors on monocytes or other phagocytic cells [31]. The only significant difference in clinical laboratory measures between PLWH who did or did not respond to TIGIT blockade with reduced TIGIT expression on their CD8^+^ T cells was a lower nadir CD4^+^ T cell count in the PLWH who did not respond. This raises the possibility that a greater extent of disease progression, even when in the past and seemingly addressed with antiretroviral therapy and subsequent immune reconstitution, may reduce the likelihood of responsiveness to TIGIT blockade. In this case, TIGIT blockade treatment would be most effective when administered before CD4^+^ T cell counts have fallen deeply. There was a wide range of TIGIT expression levels on CD8^+^ T cells of the 40 PLWH we tested, which directly correlated with age, but did not differ significantly between responders and non-responders.

The anti-CD3 redirected cytotoxicity assay we used to demonstrate that TIGIT engagement lowers the killing ability of CTL is antigen non-specific and registered relatively small reductions in general CTL activity. Since only a fraction of the CD8^+^ T cells affected in this assay are HIV-specific, we cannot extrapolate how significant or substantial the effect of TIGIT blockade on HIV-specific CTL activity would be. Given the selective expression of TIGIT on HIV-specific CD8^+^ T cells, the effect might be more substantial and in this scenario where we engage TIGIT to demonstrate inhibition, the level of inhibition observed might be decreased due to TIGIT removal by monocytes present in the PBMC [18,31]. Shock and kill strategies relying on rapid killing of HIV-infected cells following latency reversal could benefit from blockade to prevent TIGIT mediated inhibition of cytotoxicity, especially since PVR expression is upregulated on CD4^+^ T cells replicating HIV [15,21]. An assay system similar to that used to show that TIGIT blockade accelerated elimination of HIV-infected autologous CD4^+^ T cells by NK cells through antibody-dependent cell-mediated cytotoxicity could better address this possibility [21].

Although HIV-specific CD8^+^ T cell function indicated by IFN-γ production and CD107a expression was enhanced in only a minority of cases with TIGIT blockade, it demonstrates a potential benefit to immune function for at least a select subset of PLWH. We chose PLWH with strong HIV-specific T cell responses to enable response measurements by flow cytometry, but there could possibly be a greater fractional or more frequent response rate to TIGIT blockade for PLWH with low baseline T cell responses weakened in vivo due to TIGIT-mediated inhibition. The ELISpot assay we used for screening does not distinguish between CD4^+^ and CD8^+^ IFN-γ production, and we focused exclusively on CD8^+^ T cell responses by flow cytometry, potentially missing some effects of TIGIT blockade. As in cancer treatment, targeting multiple related or unrelated inhibitory receptors by mAb blockade may increase the boosting effect on antiviral immunity [32,33].

The association between trogocytosis and responsiveness to TIGIT blockade identifies monocyte status as a feature potentially relevant for selecting PLWH more likely to benefit. There is currently no consensus on the need to maintain functional Fc regions on mAb used as checkpoint inhibitors [34], but if trogocytosis is involved, the capacity for functional recognition through FcR on phagocytic cells is critical. Recent studies suggest that the CD64 FcR on monocytes plays a primary role in trogocytosis; therefore, overall monocyte frequency and the distribution of monocytes within M1 (CD64^hi^) and M2 populations may be important parameters for estimating responsiveness to checkpoint blockade [35].

Previous studies in cancer patients showed that a population of cells termed T exhaustion precursors (Tpex) are critical for robust responses to checkpoint inhibition [36,37,38,39]. These cells have limited exhaustion markers and express the TCF1 transcription factor and tissue residency chemokine receptor CX3CR1, which may associate them with the expanded population of CD5^−^CD8^+^ T cells in PLWH [40,41,42,43,44]. Further phenotypic analysis of CD8^+^ T cells rescued from inhibition by TIGIT blockade will also help predict those more likely to benefit, but the identification of monocytes as potential players illustrates how environmental context, including T cell DNAM1 expression and antigen-presenting cell PVR expression also need be considered. While we generally assume that TIGIT blockade acts by preventing delivery of inhibitory signals when TIGIT on effector cells binds to its PVR ligand, blockade can also enhance effector cell function by favoring the selective interaction of co-stimulatory DNAM-1 with PVR. Myeloid dendritic cells can also express TIGIT and release the inhibitory cytokine IL-10 following TIGIT engagement of PVR [45]. We did not collect supernatant to test whether TIGIT blockade reduced IL-10 levels in vitro or investigate TIGIT expression on myeloid DC, but we did observe enhanced responses following TIGIT blockade when there was little or no expression of TIGIT on PBMC other than on T and NK cells.

## 5. Conclusions

These results support the possibility that TIGIT blockade can improve general and HIV-specific cell-mediated immunity in a subset of PLWH. Cytotoxicity, degranulation, and cytokine release can all be reduced by TIGIT engagement on CD8^+^ T cells. Checkpoint inhibition of TIGIT should be directed to those most likely to respond, with the capacity of circulating monocytes to remove TIGIT in vitro by trogocytosis being a useful predictor and a potentially important component of responsiveness. While TIGIT blockade only enhanced CD8^+^ T cell activation when trogocytosis occurred, TIGIT trogocytosis was a weak predictor on its own. Therefore, identification of additional features underlying responsiveness to TIGIT blockade will be required to optimize its utility.

## Figures and Tables

**Figure 1 pathogens-13-01137-f001:**
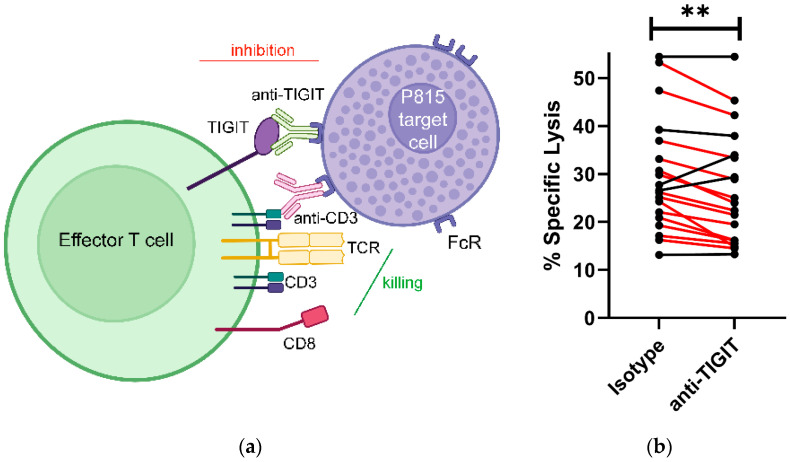
Effect of TIGIT engagement on T cell-mediated cytotoxicity. (**a**) Diagram illustrating strategy employed to enact TIGIT engagement on freshly isolated T cells triggered by anti-CD3 to lyse P815 cells (created with BioRender.com). (**b**) Comparison of anti-CD3-triggered lysis of P815 targets by PBMC from PLWH in the presence of anti-TIGIT or isotype control. Cases where specific lysis was reduced by >1/10 of baseline values by TIGIT engagement compared to the isotype control are shown with red lines and the probability of a significant reduction in specific lysis for the overall group following TIGIT engagement was calculated. (** *p* < 0.01, Student’s paired *t* test).

**Figure 2 pathogens-13-01137-f002:**
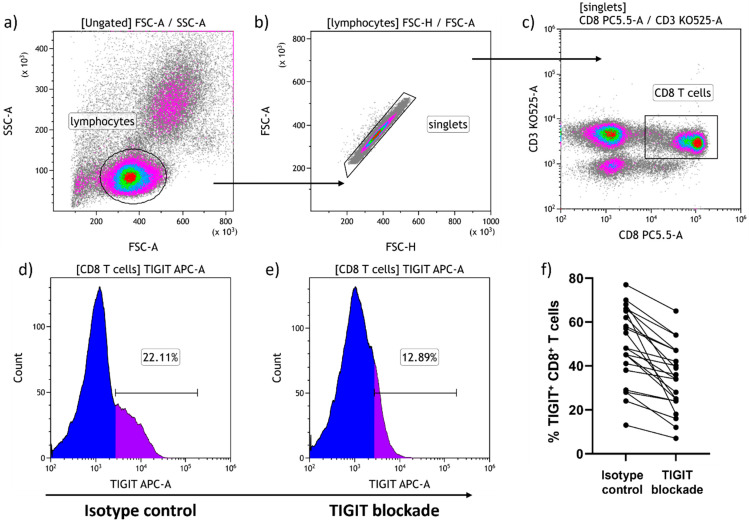
Impact of TIGIT blockade on CD8^+^ T cell TIGIT expression. To assess the effects of TIGIT blockade on HIV-specific CD8^+^ T cell function, PBMC were labeled with fluorescence conjugated anti-TIGIT mAb for 30 min before 5 h incubation with HIV Gag or Nef peptides. Additional labeled anti-TIGIT mAb was added after the 5 h incubation period with cell surface staining for CD3, CD4, and CD8. Our gating strategy for analysis of CD8^+^ T cell TIGIT expression is shown (**a**–**c**) with representative results in (**d**,**e**). Summary results for 23 subjects losing >1/10th of TIGIT expression from their CD8^+^ T cells following TIGIT blockade are shown graphically in (**f**).

**Figure 3 pathogens-13-01137-f003:**
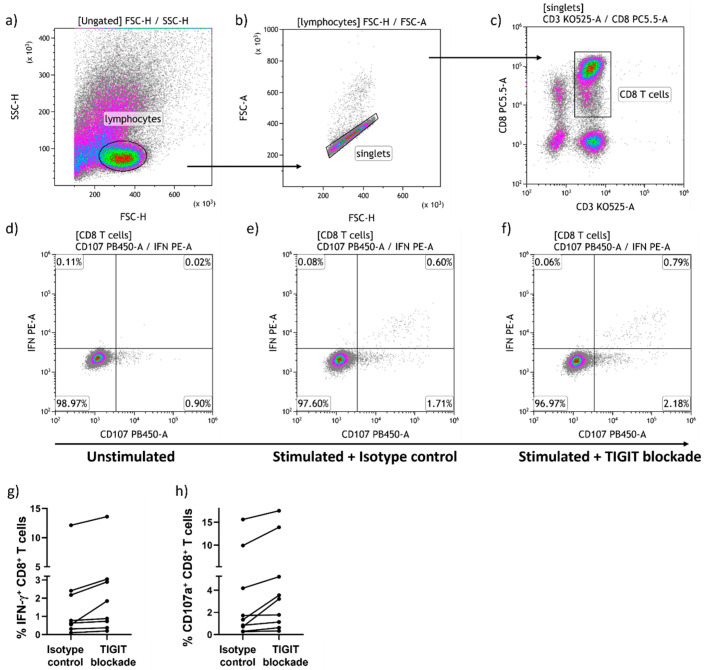
Flow cytometry analysis of HIV-specific CD8^+^ T cell activation and effect of TIGIT blockade. The gating strategy for analyzing effects of TIGIT blockade on IFN-γ production and CD107a expression by CD8^+^ T cells stimulated with HIV Gag and Nef peptides is shown with a representative example of an HIV-specific CD8^+^ T cell response enhanced by TIGIT blockade (**a**–**f**). Summary graphs of effect of TIGIT blockade on (**g**) IFN-γ production and (**h**) CD107a expression by CD8^+^ T cells from responders to TIGIT blockade with an increase >10% above that seen with isotype control treatment.

**Figure 4 pathogens-13-01137-f004:**
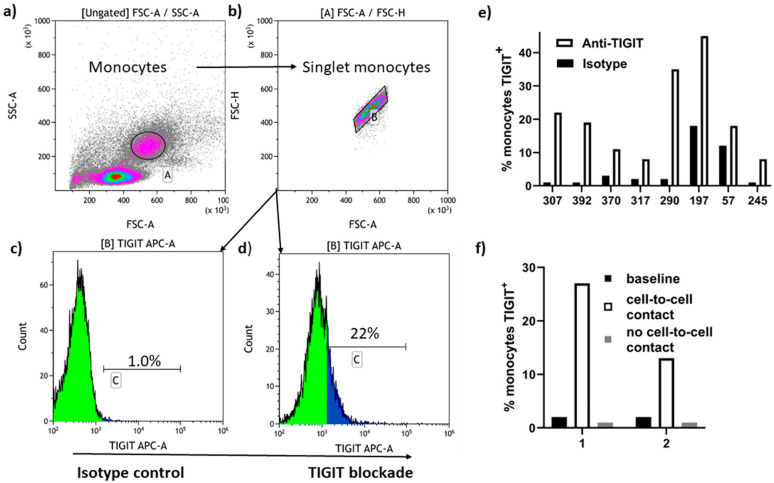
Detection of monocyte-mediated trogocytosis of TIGIT from CD8^+^ T cells. After five-hour incubation of PBMC with either fluorescent anti-TIGIT mAb or isotype control, anti-TIGIT, and isotype control-treated cells were surface stained with fluorescent anti-TIGIT mAb and monocytes gated for analysis as shown in (**a**,**b**). A representative example of the gain in anti-TIGIT mAb fluorescence on monocytes following TIGIT blockade is shown in (**c**,**d**). Letters A, B, and C within the flow cytometry plot frames refer to the gated population analyzed and the percentages above markers in (**c**,**d**) indicate the percent of cells in that gate positive for anti-TIGIT fluorescence. Monocyte identity was confirmed by 0% CD3 expression and >90% CD14 expression on cells gated in (**b**). (**e**) Summary graph of the results obtained with PBMC from eight PLWH responders to TIGIT blockade in terms of loss of CD8^+^ T cell TIGIT expression. (**f**) TIGIT blockade was carried out in PBMC separated from untreated PBMC with a semi-permeable membrane with changes in anti-TIGIT mAb fluorescence in the monocyte population shown after 5 h incubation with or without cell contact allowed.

**Table 1 pathogens-13-01137-t001:** General demographics of responders and non-responders to TIGIT blockade in terms of reduced CD8^+^ T cell TIGIT expression ^a^.

Subject	Sex	Age (Years)	CD4^+^ T Cells/μL	CD8^+^ T Cells/μL	Plasma Virus Load (log_10_ Copies RNA/mL)	Nadir CD4^+^ T Cell Count/μL	% CD8^+^ T Cells TIGIT^+^
1	male	54	1640	1680	<1.3	990	24
2	male	59	1200	740	<1.3	1200	38
3	male	56	690	1110	1.39	690	13
4	female	61	660	630	<1.3	272	33
5	male	45	ND ^b^	ND	<1.3	208	31
6	male	71	1430	1920	<1.3	27	66
7	male	49	540	1140	<1.3	540	28
8	male	61	360	660	<1.3	390	59
9	male	52	1140	1720	<1.3	464	45
10	male	57	160	460	<1.3	1	56
11	male	69	ND	ND	<1.3	480	55
12	male	57	954	2332	1.86	660	68
13	male	53	455	610	<1.3	120	48
14	male	57	1440	640	<1.3	229	48
15	male	71	740	670	<1.3	400	57
16	male	63	315	987	<1.3	513	65
17	male	57	260	890	<1.3	1	66
18	male	71	1430	1920	<1.3	27	66
19	male	61	ND	ND	<1.3	27	58
20	male	58	1045	2200	1.66	660	62
21	male	66	1010	2150	<1.3	480	70
22	male	47	800	800	<1.3	506	32
23	male	24	1400	2040	<1.3	550	24
24	male	47	1900	2550	1.56	240	77
25	female	49	980	1050	<1.3	176	32
26	female	53	480	1490	<1.3	73	73
27	male	51	323	1045	<1.3	99	46
28	male	62	1070	690	<1.3	324	31
29	male	65	760	2290	<1.3	1	56
30	male	65	320	720	2.50	170	28
31	male	69	561	816	<1.3	200	46
32	male	37	741	779	<1.3	200	30
33	female	58	580	640	<1.3	255	48
34	male	45	ND	ND	<1.3	208	31
35	male	61	240	1216	<1.3	216	58
36	male	41	940	1230	<1.3	385	41
37	male	46	520	1040	<1.3	12	39
38	male	56	1090	800	<1.3	168	45
39	male	61	550	1380	<1.3	300	69
40	male	57	470	1620	1.44	470	45

^a^ age, sex, relevant clinical laboratory measures, and baseline percentage of CD8^+^ T cells expressing TIGIT for 23 PLWH who experienced a decrease in CD8^+^ T cell TIGIT expression when their PBMC were exposed to anti-TIGIT mAb for 5 h (1–23) and 17 PLWH who did not experienced a decrease in CD8^+^ T cell TIGIT expression (24–40). ^b^ ND = not done.

## Data Availability

Data supporting the findings of this study, preserving the anonymity of study participants, are available from the corresponding author through electronic correspondence for legitimate scientific purposes.

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
