# Peer review of "Enhancement of Human Immunodeficiency Virus-Specific CD8+ T Cell Responses with TIGIT Blockade Involves Trogocytosis"

_pathogens, 2024, doi:10.3390/pathogens13121137_

Round 1
Reviewer 1 Report (Previous Reviewer 1)
Comments and Suggestions for Authors
Direct contact between lymphocytes and monocytes is needed for trogocytosis but experiments that allow or do not allow contact are lacking.
Authors do not respond to this remark.
Author Response
Direct contact between lymphocytes and monocytes is needed for trogocytosis but experiments that allow or do not allow contact are lacking.
Authors do not respond to this remark.
Thank you for pointing this out. We conducted experiments with a semi-permeable transwell insert separating PBMC treated with ant-TIGIT mAb from untreated PBMC and compared anti-TIGIT mAb fluorescence on monocytes at baseline and after 5 hours incubation. Data are presented in new figure 4f.
Reviewer 2 Report (Previous Reviewer 2)
Comments and Suggestions for Authors
I would like to thank the authors for their responses and updated manuscript.
1. It is still unclear to me what do they mean by (Lines 222-227).
Engagement of TIGIT decreased T cell killing of P815 222 target cells by more than 10% of the level observed without TIGIT engagement in 74% 223 (14/19) of the cases we tested and significantly lowered the mean specific lysis mediated 224 by the CD8+ T cells of the group (p < 0.01) (Figure 1b). These data show that in the majority 225 of cases, TIGIT engagement impairs cytotoxicity mediated by a subset of CD8+ T cells pre- 226 sent in PLWH.
Figure 1a is not showing a 10% decrease in 14/19 cases. The y axis shows % specific lysis and again, except for maybe two cases, the difference between isotype and anti-TIGIT is minimal. Can authors identify by color which 14 cases are they referring to and where is the 10% decrease?
2. My previous suggestion to modify the term “control” for “people without HIV” is to be respectful with the community. This is just a suggestion, but highly appreciated by the community.
3. Check lines 261-271 for typos (ranged from repeats)

Author Response
- It is still unclear to me what do they mean by (Lines 222-227).
Engagement of TIGIT decreased T cell killing of P815 222 target cells by more than 10% of the level observed without TIGIT engagement in 74% 223 (14/19) of the cases we tested and significantly lowered the mean specific lysis mediated 224 by the CD8+ T cells of the group (p < 0.01) (Figure 1b). These data show that in the majority 225 of cases, TIGIT engagement impairs cytotoxicity mediated by a subset of CD8+ T cells pre- 226 sent in PLWH.
Figure 1a is not showing a 10% decrease in 14/19 cases. The y axis shows % specific lysis and again, except for maybe two cases, the difference between isotype and anti-TIGIT is minimal. Can authors identify by color which 14 cases are they referring to and where is the 10% decrease?
It was unclear that we were interpreting a <10% overall change in specific lysis (eg 30-27) as a 10% change from baseline. Cases we are interpreting that way are now drawn with red lines in fig. 1b and the statistical significance of the mean change for the group with TIGIT engagement represented above the graph. Text has also been altered slightly to reflect this.
2. My previous suggestion to modify the term “control” for “people without HIV” is to be respectful with the community. This is just a suggestion, but highly appreciated by the community.
Thank you for this comment. We appreciate the community's willingness to participate in research studies such as ours and are happy use more appropriate terminology.
3. Check lines 261-271 for typos (ranged from repeats)
Typo has been corrected. Many thanks for your careful review.
Reviewer 3 Report (Previous Reviewer 3)
Comments and Suggestions for Authors
The authors have made sufficient updates and have addressed several comments.
Author Response
The authors have made sufficient updates and have addressed several comments.
Thank you for your thoughtful review.
This manuscript is a resubmission of an earlier submission. The following is a list of the peer review reports and author responses from that submission.
Round 1
Reviewer 1 Report
Comments and Suggestions for Authors
Authors aim to study whether TIGIT engagement on CD8 cells of PLHIV affects cytotoxicity so that such treatment may be useful in Cure strategies. They indeed show increased intracellular IFN staining of CD8 after TIGIT blocking, selecting cells of PLHIV that highly respond to HIV peptides in vitro. Furthermore, they provide some evidence of trogocytosis of monocytes
Questions:
1. Authors states that anti-TIGIT therapy may enhanced CD8 mediated killing and as such may be used in Cure strategies. Authors show increased intra-cellular interferon-gamma staining, however experiments showing improved killing of HIV infected cells cells are lacking. The paper would benefit if such effect is indeed demonstrated.
2. Also granzyme B and perforin produced by CD8 cells play a role in cytotoxicity: was their production capacity measured with and without anti-TIGIT treatment?
3. Trogocytosis is a fast process, so why 5 hours incubation period? Importantly, direct contact is needed and authors did not show that this process was indeed contact dependent. So how strong is evidence of trogocytosis
4. Two questions about possible mechanisms, that are nog highlighted in report:
4.1: Previous studies showed effect TIGIT on IL-10 production myeloid cells, which may be blocked by anti-TIGIT treatment. No data on IL-10 production (in supernatant) are given, although maybe relatively simple to measure in supernatant. Reduction of IL-10 mediated effects could also play a role in anti-TIGIT treatment.
4.2. TIGIT ligant is CD155. Previous studies showed that TIGIT blockade promotes CD155 binding to CD226 to activate CD8 T-cell immune activation (Chauvin et al JCI 2025). Some studies report down regulation CD155 in HIV infection, which could therefore interfere with TIGIT interventions.
Author Response
Authors aim to study whether TIGIT engagement on CD8 cells of PLHIV affects cytotoxicity so that such treatment may be useful in Cure strategies. They indeed show increased intracellular IFN staining of CD8 after TIGIT blocking, selecting cells of PLHIV that highly respond to HIV peptides in vitro. Furthermore, they provide some evidence of trogocytosis of monocytes
Questions:
- Authors states that anti-TIGIT therapy may enhanced CD8 mediated killing and as such may be used in Cure strategies. Authors show increased intra-cellular interferon-gamma staining, however experiments showing improved killing of HIV infected cells cells are lacking. The paper would benefit if such effect is indeed demonstrated.
We agree with the reviewer that showing TIGIT blockade could improve killing of HIV-infected cells would be better than showing that TIGIT engagement reduces non-specific killing by CD8+ T cells from PLWH. Unfortunately, the B lymphoblastoid cell lines commonly used as targets for HIV-specific CTL do not express the poliovirus receptor (PVR) CD155 ligand for TIGIT. To transfect autologous B cells from a substantial number of individual study subjects with PVR to allow testing of the impact of TIGIT blockade on HIV-specific killing is beyond the scope of this study. Some investigators consider CD107a expression a surrogate marker for cytotoxicity and would assume that an increase in HIV-specific CD107a induction does reflect an increase in HIV-specific cytotoxicity. We assume that the effects seen with TIGIT engagement in the non-specific redirected cytotoxicity assay would apply to HIV-specific CTL also.
- Also granzyme B and perforin produced by CD8 cells play a role in cytotoxicity: was their production capacity measured with and without anti-TIGIT treatment?
We did not measure granzyme B and perforin production with and without TIGIT blockade, only CD107a expression as a marker of degranulation. Previous studies have shown TIGIT-expressing effector cells also express granzyme B and perforin. Short term cytotoxic capacity would mostly reflect release of pre-existing stores of perforin and granzyme rather than de novo production and thus, the effect on activation as reflected in CD107a expression and interferon-gamma production would also apply to granzyme B and perforin release.
- Trogocytosis is a fast process, so why 5 hours incubation period? Importantly, direct contact is needed and authors did not show that this process was indeed contact dependent. So how strong is evidence of trogocytosis
The 5 hour time period for incubation and TIGIT blockade was chosen for T cell stimulation as per our main objective of observing an increase in cytokine production and degranulation rather than to observe trogocytosis. Testing for trogocytosis became a secondary objective after observing the decrease in TIGIT expression on CD8+ T cells with TIGIT blockade. Although a fast process, it may take time in vitro for enough cellular interaction to occur to detect an amount of TIGIT loss by CD8+ T cells and gain by monocytes measurable by flow cytometry (just as in cytotoxicity assays where killing is a fast process, but enough target cells need to be killed for a significant signal to be detected). We carried out time course studies and found that the gain of TIGIT expression by monocytes under these conditions continued through 5 hours. (New figure S2)
- Two questions about possible mechanisms, that are nog highlighted in report:
4.1: Previous studies showed effect TIGIT on IL-10 production myeloid cells, which may be blocked by anti-TIGIT treatment. No data on IL-10 production (in supernatant) are given, although maybe relatively simple to measure in supernatant. Reduction of IL-10 mediated effects could also play a role in anti-TIGIT treatment.
We did not collect supernatant to measure IL-10 production, but this is an interesting possibility that is now briefly discussed in the context of our experimental system. (lines 403-407)
4.2. TIGIT ligant is CD155. Previous studies showed that TIGIT blockade promotes CD155 binding to CD226 to activate CD8 T-cell immune activation (Chauvin et al JCI 2025). Some studies report down regulation CD155 in HIV infection, which could therefore interfere with TIGIT interventions.
We agree that TIGIT blockade may act by promoting CD226/CD155 binding and address that possibility in the discussion (lines 400-403). While CD155 may be downregulated in certain contexts in HIV infection, there are a number of reports that it is upregulated on latently and actively HIV-infected cells, in which case would TIGIT blockade would enhance activity against HIV-infected cells by preventing the TIGIT/CD155 binding that inhibits effector cell activation.
Thank you for your suggestions to improve the manuscript.
Reviewer 2 Report
Comments and Suggestions for Authors
The manuscript by Ghasemi et al, investigate the effect of TIGIT blocking on HIV-specific responses. Unfortunately, the manuscript lacks critical information to interpret the results including the participants’ demographic and clinical characteristics that are critical to discuss the variability of the responses. In addition, there is no explanation of the cut-offs selected to consider a response (including 10% and HIV-specific responses threshold). It is also extremely difficult to follow whether authors are referring to an overall 10% decrease. I would suggest modifying how the results are presented in all. In sum, in its current form it is extremely difficult to interpret the results.

I would suggest to share the manuscript with someone not related to the research to provide comments on how to clearly describe the findings. Figure legends can be improved by just detailing what the graphs mean and the statistical analysis used and leaving explanation of the results out.
Author Response
The manuscript by Ghasemi et al, investigate the effect of TIGIT blocking on HIV-specific responses. Unfortunately, the manuscript lacks critical information to interpret the results including the participants’ demographic and clinical characteristics that are critical to discuss the variability of the responses. In addition, there is no explanation of the cut-offs selected to consider a response (including 10% and HIV-specific responses threshold). It is also extremely difficult to follow whether authors are referring to an overall 10% decrease. I would suggest modifying how the results are presented in all. In sum, in its current form it is extremely difficult to interpret the results.
A description of the cohort of PLWH is completely missing. In addition, each specific assay should include a sentence containing information about the participants. Where PLWH ART-suppressed, for how long sex, ethnicity etc ? Without this information is very difficult to evaluate the manuscript. Why a 10% was increased expression of CD107a or IFN- considered significant? How was this number calculated?
We have collected the information and included a table of demographics for the 40 subjects tested in the TIGIT blockade assay and investigated differences between responders and non-responders in terms of a decrease in CD8+ T cell TIGIT expression with TIGIT blockade. Rationalization for using the 10% cut-off to categorize responders is offered.
I suggest modifying the term non-HIV infected controls using the most up to date NIH guidelines. One possibility could be people without HIV.
We have substituted the term controls for non-HIV infected controls whenever used.
Figure 1. Lines 215-218. Interpretation of these results are confusing. There is an overall decrease of the % of specific lysis. However, based on the figure, except maybe for two samples that achieved >10% decreased lysis with anti-TIGIT, most of them had very little difference compared to isotype, and responses were variable. In addition, the 10% cut-off is not justified.
We only tested individuals who had >10% baseline specific lysis as that is conventionally considered the minimum level of cytotoxicity above background demonstrating specific killing. While results with a higher level of killing are more reliable and convincing, we have used a 10% change in baseline levels of cytotoxicity in duplicate tests previously as an effect threshold.
Figure 2. This figure should be supplemental.
Moved to supplemental data as suggested.
Llines 234-235, how and why did the authors choose 0.1% of total PBMCs?
Although instrument specifications may have improved recently, we followed previous flow cytometry guidelines stating that accuracy and precision require acquisition of a minimum number of events above instrument background. 0.1% of 106 PBMC is 1000 events and a 10% change in the response would involve 100 events, which is well above the instrument background.
The lack of the cohort characteristics precludes interpretation of the results (lines 248- 249), the broad variability could be due to differences on participants clinical/demographical characteristics.
We have collected cohort characteristics and investigated differences between responders and responders. Only nadir CD4+ T cell count and correlation between CD8+ T cell TIGIT expression and age differed between the groups.
Figure 3. CD8 gating seems to be a little off. Did authors check and exclude double positive CD4+CD8+ cells? Do CD8 intermediate/low cells express TIGIT as well? What is the contribution of this population when TIGIT is blocked? The arrow below d) and e) is not in the correct position (either TIGIT-APC one time for both or no arrow at all). Again, the 10% difference explanation is not clear. There are certainly participants with less than 10% difference. Lines 269-270, figure legend: summary of results for 23 subjects losing >10% TIGIT from their CD8+ T cells following TIGIT blockade is demonstrated in (f). Do authors refer to an overall >10%?
On what is now figure 2, we followed the gating strategy as shown, first on lymphocytes, then on CD3+CD8+ lymphocytes. Very few, if any, CD4+CD8+ double positive T cells are expected in the CD3+CD8+ quadrant. Those that are, we are including as CD8+ T cells. Although those that do not express CD3 are natural killer cells, CD8+ intermediate/low cells do express TIGIT and produce cytokines when stimulated appropriately. We did not examine CD8hi and CD8lo cells separately in response to TIGIT blockade. I’m not sure what the issue is with the TIGIT-APC label below d) and e) as the X-axis for both plots measures TIGIT-APC fluorescence. The loss of 10% TIGIT expression refers to fractional loss of TIGIT expression, not absolute. We have attempted to clarify this point whenever possible in the methods and results sections.
Figure 4 is completely unclear and again the main problem is the lack of a clear explanation of the number and characteristics of the cohort included for that particular experiment. That coupled with the lack of clarity of the 10% cut off makes this figure almost impossible to interpret.
What is now figure 3 shows a single representative example illustrating our flow cytometry gating strategy for analysis of the effect of TIGIT blockade on IFN-g production and CD107a expression following stimulation of PBMC from PLWH with HIV Gag and Nef peptides. Results for responders to TIGIT blockade are summarized graphically in g) and h).
I would suggest to share the manuscript with someone not related to the research to provide comments on how to clearly describe the findings. Figure legends can be improved by just detailing what the graphs mean and the statistical analysis used and leaving explanation of the results out.
Thank you for your helpful and constructive comments. We asked several scientists not involved to read the revised manuscript, comment on what was unclear and provide suggestions to improve clarity.
Reviewer 3 Report
Comments and Suggestions for Authors
The paper talks about the possibility of TIGIT blockade and its potential role in HIV-specific cell mediated immunity in PLWH. Authors mention this as a prognostic marker for identifying patient population which can benefit from TIGIT engagement.
The reviewer has the following queries/suggestions to this manuscript:
Title: Check the spelling of Human
Line 19: Please expand the acronym PLWH at first instance.
Line 216: Although significant, is this specific lysis physiologically relevant?
Line 255: What about the other 17/40 PLWH patients? What is the base level TIGIT expression in those patients? Did the loss of TIGIT expression of the 23 PLWH population correlate with IFN gamma production in Fig.2?
Line 281: Authors mention 23 patients showed >10% loss of surface TIGIT from CD8+ population. In figure 4, they only show 6/23 PLWH having enhanced IFN gamma and CD107a upon TIGIT blockade. What about the other 17/23 patients who showed >10% loss in surface TIGIT expression from CD8+ T cells?
Fig 4g/4h: Is this data significant? What is the therapeutic relevance of this increase?
Line 304-306: Have the authors tested internalization/subsequent degradation of surface TIGIT by the fluorescent anti-TIGIT antibody?
Line 306-309: Monocytes also express surface TIGIT. Why couldn't the fluorescent antibody be taken up by monocytes instead of Trogocytosis?
Comments on the Quality of English LanguageThe quality of English is good (except for the typo in the title).
Author Response
The paper talks about the possibility of TIGIT blockade and its potential role in HIV-specific cell mediated immunity in PLWH. Authors mention this as a prognostic marker for identifying patient population which can benefit from TIGIT engagement.
The reviewer has the following queries/suggestions to this manuscript:
Title: Check the spelling of Human
The spelling of Human has been corrected.
Line 19: Please expand the acronym PLWH at first instance.
Abbreviation PLWH spelled out on first use in abstract and introduction.
Line 216: Although significant, is this specific lysis physiologically relevant?
We discussed this (lines 366-374) and why we believe it is likely to be physiologically relevant.
Line 255: What about the other 17/40 PLWH patients? What is the base level TIGIT expression in those patients? Did the loss of TIGIT expression of the 23 PLWH population correlate with IFN gamma production in Fig.2?
The base levels of TIGIT expression in all 40 PLWH selected for study distinguished as responders or non-responders are now shown in Table 1. There was no significant correlation between the loss of TIGIT expression and IFN-gamma production.
Line 281: Authors mention 23 patients showed >10% loss of surface TIGIT from CD8+ population. In figure 4, they only show 6/23 PLWH having enhanced IFN gamma and CD107a upon TIGIT blockade. What about the other 17/23 patients who showed >10% loss in surface TIGIT expression from CD8+ T cells?
Even though they lost >10% surface TIGIT expression from their CD8+ T cells with TIGIT blockade, these 17 PLWH did not have enhanced IFN-gamma and CD107a expression. Two of the 17 had enhanced IFN-gamma or CD107a expression. We discuss possible reasons for this (lines 357-363).
Fig 4g/4h: Is this data significant? What is the therapeutic relevance of this increase?
We chose to represent those subjects where there was an increase in IFN-gamma production and CD107a expression on these graphs, therefore, the statistical significance of the difference is not meaningful. The therapeutic relevance of this increase is theoretically meaningful in that the antiviral effector functions of HIV-specific CD8+ T cells are enhanced, but this has not been empirically demonstrated.
Line 304-306: Have the authors tested internalization/subsequent degradation of surface TIGIT by the fluorescent anti-TIGIT antibody?
We did not directly test for internalization and degradation of surface TIGIT by the fluorescent anti-TIGIT antibody, but if internalized, the fluorescent signal would be maintained by the CD8+ T cell.
Line 306-309: Monocytes also express surface TIGIT. Why couldn't the fluorescent antibody be taken up by monocytes instead of Trogocytosis?
We accounted for baseline TIGIT expression by the monocytes with the control anti-TIGIT staining of monocytes after 5 hours with an isotype control added to the cultures versus the level of anti-TIGIT staining when anti-TIGIT antibody was added to the cultures for 5 hours. The increase seen we assume occurs through trogocytosis (as it is also disappearing from the CD8+ T cells) and not de novo expression by the monocytes. In most cases, there is very little expression of TIGIT in the monocyte population.
Thank you for your constructive and helpful suggestions.